# Transducer Technologies for Biosensors and Their Wearable Applications

**DOI:** 10.3390/bios12060385

**Published:** 2022-06-02

**Authors:** Emre Ozan Polat, M. Mustafa Cetin, Ahmet Fatih Tabak, Ebru Bilget Güven, Bengü Özuğur Uysal, Taner Arsan, Anas Kabbani, Houmeme Hamed, Sümeyye Berfin Gül

**Affiliations:** Faculty of Engineering and Natural Sciences, Kadir Has University, Cibali, Istanbul 34083, Turkey; mustafa.cetin@khas.edu.tr (M.M.C.); ahmetfatih.tabak@khas.edu.tr (A.F.T.); ebru.bguven@khas.edu.tr (E.B.G.); bozugur@khas.edu.tr (B.Ö.U.); arsan@khas.edu.tr (T.A.); anas.kabbani@stu.khas.edu.tr (A.K.); houmemehamed@stu.khas.edu.tr (H.H.); 20171709010@stu.khas.edu.tr (S.B.G.)

**Keywords:** transducers, biosensors, wearables, analytes, bioreceptors

## Abstract

The development of new biosensor technologies and their active use as wearable devices have offered mobility and flexibility to conventional western medicine and personal fitness tracking. In the development of biosensors, transducers stand out as the main elements converting the signals sourced from a biological event into a detectable output. Combined with the suitable bio-receptors and the miniaturization of readout electronics, the functionality and design of the transducers play a key role in the construction of wearable devices for personal health control. Ever-growing research and industrial interest in new transducer technologies for point-of-care (POC) and wearable bio-detection have gained tremendous acceleration by the pandemic-induced digital health transformation. In this article, we provide a comprehensive review of transducers for biosensors and their wearable applications that empower users for the active tracking of biomarkers and personal health parameters.

## 1. Introduction

Living in the world of information technologies, we interact with sensors and transducers daily through smartphones, wearables, cars, cities, homes, and offices. After decades of development and commercialization phases, sensor-and- transducer-based systems have been greatly improved and have facilitated a more comfortable daily life. Today’s commercially available sensors use a wide spectrum of signal conversion mechanisms for providing instantaneous feedback on personal health, environmental conditions, events, and changes. With the help of the transducer integrated systems, applications such as autonomous driving, robotics, and smart homes are already making people’s lives easier, more comfortable, and safer.

In parallel to the foreseeable progress of sensor and transducer technologies, complementary connection technologies, such as the internet of things (IoT) and 5G, pave the way toward larger-scale systems, leading to telehealth, smart cities, and autonomous transportation. Biosensors in health and fitness wearables play a crucial role by extracting the bio-information required as an input for the abovementioned emerging applications. To that end, wearables aim to provide low-cost, instant feedback on biomarkers using efficient signal conversion mechanisms. In the development of wearable biosensors, the signal conversion mechanism simply defines the functionality and compatibility of operation on human skin. Therefore, transducers define the wearable form factor and potential user adoption of a particular device as the main elements of signal conversion. In this review article, we will focus on the transducer technologies for biosensors and their wearable applications to present a concise outlook on the prospects and challenges, as well as the basics of biosensing for a complete understanding of the concept.

Starting from the first ‘‘biosensor’’ demonstration by Clark and Lyons in 1962 [1], in which the preparation of an enzyme electrode provided the transformation of glucose into a detectable current output via oxygen reduction, biosensors took decisive steps in bio-analysis, such as potentiometric urea electrodes by Guilbault and Montalvo in 1969 [2], and the first commercial biosensor for glucose detection in 1975 [3]. Today, biosensors have increasingly come to define an entire realm of commercial applications that encourage a proactive approach to preserve one’s health so as to prevent illness in lieu of battling it after the fact. The impact of biosensors has been promoted with their existence in wearables that open a venue for research and industry to supply the demand of increasing standards of living. To that end, the development of clinical-grade biosensors and their wearable device applications are expected to give predictive guidance for clinical interventions. With the achievement of clinical-grade accuracy and precision in health and fitness wearables, personal vital sign extraction might become a prerequisite before requesting an appointment from a health professional.

Biosensors refer to the collaboration of receptors that recognize target analytes and transducers that translate this recognition into a detectable signal [4]. Biological molecules such as enzymes, nucleic acids, antibodies, or their synthetic analogues can serve as bio-receptors to bind the analyte of interest. To form a biosensor device that detects or measures the biological events or changes, the targeted matching of the bio-receptor and the analyte should be evaluated quantitatively, making the transducers indispensable components of a biosensor [5,6]. Availability of various bio-receptors, transducers, and possible combinations of both components constitute various ways to classify biosensors (Figure 1).

Biosensors can monitor target analytes such as biomarkers, pathogens, or allergens. All can be exploited as an indicator of the health status not only to diagnose but also to monitor the patient’s prognosis [7,8,9,10,11]. This track record is favored, especially for elderly individuals and individuals with alcohol or drug abuse cases [12,13]. Monitoring the levels of exogenous substances is a leverage for the healthcare professionals to provide full-fledged guidance.

The analytes are the biological components of a biosensor that can be extracted from different bodily fluids such as sweat, saliva, urine, and blood (Figure 1). In addition to bio-recognition from bodily fluids, the variety and extended functionality of transducing mechanisms allow for the extraction of vital signs such as basal body temperature (BTT), heart and respiratory rates, systolic and diastolic pressures, and even tremors. When combined with the readout electronics, the output signal from the transducers can be traced, analyzed, and recorded to evaluate the health-related quality of life (HRQOL) [5,14,15,16,17]. In today’s world, the maintenance of HRQOL is raising concern in society, which increases the demand for wearable biosensors monitoring vital processes. To that end, wearables have brought a completely different perspective to modern medicine from fitness trackers to medical devices in the clinical setting.

Technologies offering mobility to patients and clinicians authenticated the contribution of wearable biosensors in remote detection and the monitoring of individuals’ health status. Perceiving the signs of potential clinical issues in advance has improved the maneuverability of health care professionals not only to monitor their self-quarantined patients but also to increase their preparedness by tracking their physiological status as front-line workers [18,19].

Determining and recording the physiological parameters and comparing them to the critical thresholds with absolute precision has gradually become easier over the years with the foreseeable progress of biotechnology. The bio-analytical systems have radically progressed from the times when samples were taken from the relevant person and delivered to different laboratories to the point-of-care (POC) diagnostics, a bedside patient follow-up unit, which is directly accessible to the person concerned [5,20]. In line with this progress, the continuous tracking of body outputs is now provided by wearable biosensors [5,21]. The breakthrough advantages of wearable technology compared to the conventional bio-analytical methods or POC testing devices are that their continuous monitoring does not require an invasive way to collect samples from the person of interest and can be performed in a user-friendly operation at a low cost [22,23,24,25]. The academic and industrial interest in wearable technologies greatly attracted the development of new mobile devices advancing biosensors by combining them with new materials and compact electronics. Based on our current search on PubMed for the articles published with the keywords “wearable” and “biosensor” in their abstracts, we can deduce that nearly half (47%) of those address the significance of the predictive and personalized remote advantages. Likewise, wearable biosensors specific to the diagnosis and prognosis of COVID-19 have strengthened their use in the market for healthy individuals, as well as for patients during the pandemic [26]. According to Researchandmarkets’ report, the global market of wearable technologies holds a value of USD 47.89 billion with expected growth reaching the value of USD 118.16 billion in 2028 [27]. Owing to advantages such as fast response, specificity, and sensitivity, commercially developed wearable biosensors for the medical industry have gone through a radical shift for self-testing at home and critical care at the bedside in emergencies [28]. Furthermore, the digital health transformation accelerated by the pandemic strongly depends on wearable biosensor technologies to provide accurate and continuous sensing of physiological information [29].

Playing a crucial role in signal conversion for biosensing, transducers provide output quantity with a given relationship to input quantity [30]. The compatibility of biosensing technologies to the wearable form factors strongly depends on the transducing technologies, therefore, advancements in the transducers and the miniaturization of readout electronics with wireless data communication technologies created stark improvements in the field of wearable biosensing, especially in the consumer-based products that are widely available in the market. In the next sections, we will focus on the construction of transducers by giving insight into their signal conversion mechanisms, and provide a detailed classification of biosensor technologies based on such mechanisms.

## 2. Construction and Classification of Transducers for Biosensors and Wearables

Biosensors have been evolved as the combination of bio-receptors and transducers based on electrochemical, optical, thermal, and gravimetric methods converting the signals from analytes containing antibodies, nucleic acids, and immunological agents, microorganisms, hormones, enzymes, cells, tissues, chemical receptors, and other detectable biological inputs. For most biosensors, device construction entails three steps; (i) implementation of a bio-receptor that reacts with a specific analyte, (ii) integration of a transducer, and (iii) the fixation/immobilization of a biological component to the transducer. Therefore, creating a biosensing device strongly depends on these construction steps together with device design strategies and integration of readout electronics for wearable devices that will provide continuous use on the human body. It is also important to acknowledge that the isolation from external factors such as chemical/physical conditions (temperature, contaminants, and pH), should be further taken into consideration while constructing wearable biosensors for specific applications [31,32].

The variety of available transducing mechanisms provides wearable biosensors worn on the head, neck, torso, legs, feet, arms, hands, and fingers. Figure 2A shows the body locations for which the multiple wearable form factors have been reported [33,34,35]. The wearable market and reported research results include a broad spectrum of device designs from smart helmets to skin patches (Figure 2A), for which user adoption and accuracy are the critical factors to present a sustainable and user-friendly technology. To that end, the construction of transducing mechanism is a key factor that can convert many biosensing technologies to be used as wearable devices.

Transducers for biosensors depend on the type of material used, specifications of the sensor device, and the actual signal conversion mechanism (Figure 2B). The transducer materials are generally classified as inorganic, organic, conductor, insulator, and semiconductor, and can also be found in the form of a biological substance. While the specifications of the transducer are mostly defined by the capabilities of the active sensing material, the design of the device also plays an important role in the definition of the final specifications (Figure 2B). To that end, transducer mechanism simply defines the class of the biosensors, for instance, a biosensor is classified as an “electrochemical biosensor” if it uses an electrochemical transducer. Following the conventional classification, from now on, we will classify the biosensors depending on their transducers.

### 2.1. Electrochemical Biosensors

Electrochemical biosensors were the first scientifically proposed and successfully commercialized biosensors for multiple analytes [1,37,38,39,40]. Electrochemical biosensors are constructed with bio-analytes and electrochemical transducers. They utilize chemical reactions between the immobilized biomolecule and target analyte that produce/consume ions or electrons that affect the measurable electrical properties (e.g., electric current or potential) of the solution [37,39,41]. These biosensors are primarily based on Faraday’s laws of electrolysis and Faradaic current resulting from the direct transfer of electrons in redox reactions at the heterogeneous electrode-solution interface [42]. In such redox reactions, reference electrodes, such as silver tetramethylbis (benzimidazolium) diiodide [43], must be held still at a fixed potential so that they neither affect the working electrodes nor be affected by the solution. It is, however, highly important to develop reusable and miniaturized reference electrodes as the commercial demand has substantially risen for electrochemical biosensing-based technologies [44].

The reported advantages of electrochemical biosensors include ease of use, better signal-to-noise ratios with lower background noise, ability to operate on small sample volumes, cost-effectiveness in production, and compatibility to downscaling together with the minimal power requirements on the device operation [40,45,46]. On the other hand, reported limitations are (i) the unreliable dynamic range of measurement due to enzyme saturation kinetics, (ii) potential interference of other compounds in solutions, (iii) oxygen requirement and concentration fluctuation in solution, particularly for glucose measurement, (iv) the influence of pH values or ionic forces on the enzymatic activities, and (v) biochemical and slow electron transfer processes restricting the efficiency and speed [40].

Electrochemical biosensors can be further categorized into three main types: amperometric/voltammetric, potentiometric/conductometric, and impedimetric/capacitive biosensors. All these different types of electrochemical biosensors serve in a wide spectrum of applications ranging from detecting pollutants and pathogens to clinical diagnosis of various diseases [47], such as sensors for glucose and H_2_O_2_ monitoring [48,49], immunosensors [50,51,52] for identification of several viruses, such as plum pox [53], fig mosaic [54], and avian leukosis subgroup J [55]. In addition, agricultural, environmental, and industrial applications have also been reported by using electrochemically transducing biosensors [56].

#### 2.1.1. Amperometric and Voltammetric Biosensors

Amperometric biosensors continuously measure the current $\left(i\right)$ or the current density $\left(j=\frac{i}{A}\right)$ (per unit area A), that can be generated through oxidation or reduction by a biochemical reaction at the surface of a working electrode [40,57,58,59]. The electrode species can be made up of graphite, noble metals, and modified forms of carbon or conducting polymers [60,61,62]. The simplest form of amperometric biosensing is the direct current (DC) measurements, which can be well-represented by their ancestor Clark oxygen electrodes [1].

Amperometric biosensors that provide linear concentration dependence over a defined range [39] offer advantages of rapid robustness, portability, high sensitivity, low cost, and low limit of detection [46,63]. However, in order to promote the electrochemical reaction of a selected analyte at the working electrode, suitable analytes must be introduced. Some analytes, such as protein-based ones, may not be able to serve as good redox partners in such reactions [64]. It is also reported that, due to the uncompensated intrinsic resistance of the DC measurements, the specificity and selectivity of this technique may be insufficient for applications requiring high sensitivity [59].

Amperometric transducers have been reported to be implemented in wearable biosensors successively [65,66]. By using amperometric transducing, Kim et al. have demonstrated a wearable mouthguard with integrated wireless data communication [67]. Figure 3A shows the demonstrated wearable mouthguard and the placement of the receptor and the transducer accordingly. In this device, the authors used a wireless amperometric circuit to detect the uric acid (UA) directly from the saliva. The UA is considered a crucial biomarker for the detection of various diseases such as Lesch–Nyhan and Renal syndrome; therefore, wearable amperometric transducing with the saliva as an input is a suitable and promising method for non-invasive and low-cost monitoring of metabolites [67]. To that end, the authors used screen printing to form the working electrodes on PET (polyethylene terephthalate) substrate and fabricated the Prussian-blue transducer (Figure 3B) by crosslinking the uricase enzyme and electropolymerizing *o*-phenylenediamine. To test the amperometric mouthguard, the undiluted human saliva samples were first obtained from volunteers and the concentration of the saliva samples was determined up to the artificial saliva. The authors have recorded the changes in the current output with the application of −0.3 V bias voltage (Figure 3C). The data is wirelessly communicated to the final device through the fabricated printed circuit board (PCB) and the programmable Bluetooth low energy (BLE) module. The inset of Figure 3C shows the linear calibration data extracted from the amperometric changes. Figure 3D shows the stability of the demonstrated wearable sensor through repetitive measurements. Kim et al. have demonstrated that the amperometric wearable sensor has provided good linearity and stability that holds promise for continuous monitoring. The inset of Figure 3D shows the percentage changes of the original current response due to instabilities.

The amperometric biosensing subclass also includes voltammetry, in which the current is measured during the controlled variations of the potential [70]. Promising features of voltammetry opened significant opportunities in electrochemically transducing biosensing systems, through voltammetric biosensors. Voltammetry is an electro-analytical method, from which information is obtained for an analyte by varying potential, and then, the resulting current can be measured accordingly. There are a variety of forms of voltammetry sourced from the alternative ways of potential modulation, such as polarography [71,72], cyclic voltammetry (CV), normal pulse, reverse pulse, linear sweep, differential staircase and differential pulse, and square wave [73,74,75]. Of these, CV is one of the most widely used techniques that provides information about redox potential and electrochemical reaction rates of analyte solutions. The working principle of voltammetric biosensors simply utilizes the change in current as a function of the varying potential. This principle is required to detect analytes in solutions, in which the peak current value provides information for identification, while peak current density is proportional to the concentration of the corresponding species. The advantages of voltammetric biosensors are high sensitivity in measurements and simultaneous detection of multiple analytes [76]. Considerable logistic factors and features have been reported, such as the mechanical strength, cost-efficiency, availability, stability, and easy alteration of structures. The construction of such biosensors can be made up of various materials, such as blue-dendrimer nanocomposites [77], glassy carbon, polycrystalline boron-doped diamond, carbon nanotubes [78], graphene, and carbon-paste electrodes (CPEs) with multi-walled carbon nanotubes, CPEs with graphite, or carbon microspheres [79,80].

#### 2.1.2. Potentiometric and Conductometric Biosensors

Biosensors using potentiometric transducers measure the difference in potential (oxidation or reduction potential) generated across an ion-selective membrane separating two solutions at virtually zero-current flow for ranging analyte activities. The working principle of potentiometric biosensors relies on a current flow occurring in an electrochemical reaction, where a ramp voltage is applied to the electrodes in the solution. Potentiometric transduction operates through a ‘‘Boltzmann distribution applied to the redox-active molecular systems’’, namely, the Nernst Potential in a simple form [59,61]:$$V=V^{0}+\frac{k_{B}T}{e}\ln\frac{\left[Ox\right]}{\left[Red\right]}$$
where V denotes the electrode potential, $V^{0}$ denotes the formal potential of redox probes, *T* is absolute temperature, $k_{B}$ is the Boltzmann constant, *e* is the charge of an electron, and the ratio $\frac{\left[Ox\right]}{\left[Red\right]}$ is the concentration ratio of oxidized and reduced species. Basically, potentiometric biosensors deal with the potential difference $\left(V-V^{0}\right)$ in the above equation due to a biological event that changes the equilibrium of the redox reaction. Therefore, by definition, it can be written as a change in the Gibbs free energy [60]:$$\Delta G=e\left(V-V^{0}\right)=k_{B}T\ln\frac{\left[Ox\right]}{\left[Red\right]}$$

Potentiometric biosensors offer compatibility with downscaling and modern silicon fabrication technologies [81]. Many of them are commercially available, such as pH electrodes, ion-selective electrodes (ISEs), glass electrodes, and metal oxide-based sensors. While ion-selective membranes of ions, (H^+^, F^−^, I^−^, Cl^−^), and gases (CO_2_, and NH_3_) are currently available for biosensing, researchers focus more on creating novel membrane compositions with carbon-based materials, polyvinyl chloride (PVC), and unique ionophores [82] to enhance the capability and uses. Such membranes are reported to be primarily used in ISEs and ion-sensitive field-effect transistors (ISFETs) [37,39,40].

ISFET technology is also classified under potentiometric biosensors similar to the ISEs. ISFETs are reliable, well-applicable, analytical devices, and have many advantages in biosensors, such as suitability for mass production, easy production of small-sized devices with the semiconductor manufacturing process, robustness, applicability of non-conductive materials, and fast response. ISFET is a classical metal oxide semiconductor FET with a gate formed by a separated reference electrode and attached to the gate area via an aqueous solution [83,84].

ISFETs have an ion-sensitive surface. When any interaction between the semiconductor and ions occurs, the surface electrical potential changes, which can subsequently be measured. A selectively permeable polymer layer, from which ions diffuse through and change surface potential, can also be used to cover the sensor electrode in the construction of the ISFET (also called Enzyme-FET or ENFET, having very low detection limits but allowing the use of small sample volumes) [84,85]. Different types of ISFET biosensors have been reported, such as ultra-thin body dual-gate ISFET, silicon nanowire ISFET, and biologically sensitive field-effect transistors. Common drawbacks of the ISFETs have been reported as high dependence on pH changes and insufficient measurement capabilities for blood and blood plasma samples with high buffering capacity [86,87,88].

The incorporation of the potentiometric transducer in wearable technologies has also been frequently reported [89,90,91]. By using a potentiometric transducer mechanism, Gao et al. demonstrated a fully integrated wearable sensor array for perspiration analysis [68]. As a potentiometric transducer, the authors have used PEDOT:PSS (poly(3,4-ethylenedioxythiophene) polystyrene sulfonate) in the ISEs; and carbon nanotubes in the polyvinyl butyral (PVB) reference membrane. This way, they created mechanically robust potentiometric transducers that can be worn on the subject’s wrist or head for sweat analysis (Figure 3E). The demonstrated wearable array of transducers includes potentiometric ISEs for Na^+^ and K^+^ ions and amperometric transducers for glucose and lactate sensors with a complementary resistance-based temperature sensor (Figure 3F). The integrated sensor array in this work allows multiple transduction mechanisms so that the simultaneous measurement of glucose and lactate is through amperometric transduction, while the detection of Na^+^ and K^+^ ions is potentiometric. The authors have demonstrated the experimental characterization of potentiometric transduction to sense Na^+^ and K^+^ in NaCl and KCl solutions, respectively, and further demonstrated the real-time sweat analysis from a subject wearing a headband device during stationary cycling (Figure 3G,H). To provide simultaneous real-time measurements of four different analytes from the sweat and the complementary resistance-based temperature sensing, the authors used a programmable microcontroller integrated with a Bluetooth module on a PCB for the data processing and wireless data communication to a mobile phone. As shown in this study, mechanically flexible wearables and the integration of compact electronics into them enhance potentiometric and amperometric transducers in wearable forms yielding real-time detection of biomarkers in indoor and outdoor activities.

Conductometric transducers for biosensors can also be classified in the subgroup of potentiometric transducers. Conductometry aims to measure the change in the conductivity of ionic species with respect to the bio-recognition event. Conductometric biosensors have been demonstrated to sense a variety of analytes such as glucose urea and arginine [92]. Even though they have specific advantages, such as the operation without the need for reference electrodes, compatibility to operate at low-amplitude alternating voltages (which prevents Faraday processes on electrodes), insensitivity to light, and easy integration and miniaturization, the wearable biosensing applications of conductometric transducing methods have been suppressed by the limitations such as higher signal-to-noise ratio (>2%) causing lower sensitivity, low specificity resulting in the incapability of distinguishing between simultaneous reactions, and occurrence of polarization in the electrodes of the double-layer capacitance during the reaction. In addition, the response value of such biosensors highly depends on medium conditions, such as pH and ionic strength, and buffer capacity [93]. Thus, the selectivity of the conductometric method is presumed to be low and, consequently, its potential use for wearable different applications encounters technical difficulties.

#### 2.1.3. Impedimetric and Capacitive Biosensors

Impedimetric biosensors depend on the impedance measurements resulting from the redox reactions in the analyte/electrode interface due to a biological recognition. This technique includes the application of a small perturbation bias to sense the change in the oscillations of the current response. Impedimetric measurements require a definition of a mathematical transfer function for the electrochemical impedance, which is a complex function of frequency denoted by $\left[Z^{*}\left(\omega \right)\right]$. In the simplest form, the complex impedance function is sourced from the ratio of the perturbed voltage with a frequency of ω,
$$V\left(t\right)=V_{0}+V_{p}e^{i\omega t}$$
to the output current response with a phase difference of ∅,
$$I\left(t\right)=I_{0}+I_{p}e^{i\omega t-\emptyset }$$
yielding,
$$\left[Z^{*}\left(\omega \right)\right]=\frac{V\left(t\right)}{I\left(t\right)}=Z+iZ^{\prime }$$
where Z and $Z^{\prime }$ are the real and complex parts of the impedance, respectively. The impedimetric biosensors have been reported for the detection of bacteria and whole cells [94] Moreover, impedimetric transducing yields the electrochemical impedance spectroscopy (EIS) technique, which is a highly used biosensing technology based on scanning over the perturbation frequency to measure the resulting impedance changes in the interface of bio-receptor and analyte [95]. Impedimetric transducing has also been incorporated into wearables. Lee et al. have demonstrated a wearable device in the form of a semi-transparent and flexible skin patch using impedimetric transducers for the active monitoring of glucose in diabetic patients [69]. The demonstrated wearable device consists of multiple sensors for humidity, glucose, pH, and tremor detection (Figure 3I). To provide a mechanically stable and optically transparent electrochemical interface, the authors used a serpentine mesh of gold (Au) and Au-doped graphene, yielding a stable transfer of signals. The device uses a sweat uptake layer, sensing components, and therapeutic components such as microneedles, a heater, and a temperature sensor to release the drug above the threshold skin temperature. The authors demonstrated multiple transducing of their device, including voltammetric, and impedimetric transducing by using Au film, Au mesh, and Au-coated graphene (Figure 3J–L). All three electrode structures are tested in phosphate-buffered saline with Fe(CN)_6_^3−/4−^ to provide voltammetric (CV) (Figure 3J), and impedimetric (Figure 3K,L) responses. The authors further demonstrated the stability of the transducing mechanisms under the application of mechanical stress and applied the wearable patch to a diabetic mouse to prove the successful drug release correlated with the sensor operation. The device is finally reported to be attached to healthy individuals from whom the glucose and pH measurements are taken. The device shows a statistically high level of correlation factor (*p* < 0.001, R^2^ = 0.89) to the commercially available glucose assay kit for sweat and reliability to the blood glucose meter measurement.

Capacitive transducing is another promising method for bio-detection and has been widely investigated for potential integration into wearables for human health monitoring. In a biosensor that uses capacitive transducing (capacitive biosensor), an analyte binds by interacting with bio-receptors grafted or immobilized on the electrode surface [96]. The capacitive transducers are demonstrated to present compatibility in the detection of analytes, e.g., hormones or DNA fragments, by registering the signal changes due to the bio-events modulating the dielectric properties or thickness of the immobilized sensing layer [97]. In this regard, the development of affinity-based capacitive biosensors was first demonstrated in the 1980s depending on changes in dielectric properties, dimension, shape, and charge distribution, when an antibody/antigen complex formed on the surface of an electrode [98].

Analog to the parallel plate capacitors, the active surface of capacitive biosensors serves as one of the plates of the parallel-plate capacitor which detects changes in capacitance that can be expressed as:*C* = *ε·A*/*d*
where *ε* is the dielectric medium permittivity, *A* is the area of plates, and *d* is the distance between plates. If the capacitance between the plates is needed to vary, one can simply change the *d*, *ε*, or *A*. Preferably, the capacitance can be measured using a bridge circuit, where the output of transducer impedance is given by:*X_C_* = 1/2π*f·C*
where *C* is the capacitance, and *f* is the excitation frequency.

In general, affinity-based capacitive biosensors can detect both conductive and non-conductive target samples, which can operate independently of the readout distance due to a change in the electric field around it [99]. The second plate in the capacitor analogy is the analyte to be detected for the conductive target samples. On the other hand, to measure non-conductive target samples, a metallic plate serves as the second plate and the target sample is the insulator between the parallel plates. Since a capacitive sensor measures the change in dielectric properties and thickness of the dielectric layer, the precision of the capacitive biosensor varies according to parameters such as the concentration of charged ions at the electrode–analyte interface, the distance of the electrode plates, and the content of the analyte. Capacitive transducers provide label-free detection as they utilize a method based on measuring changes upon binding of the analyte to a ligand/receptor immobilized on the electrode surface [100]. Measurements can be directly performed in real-time, without dependence on expensive labels. In this respect, biosensors using capacitive transducers have novel advantages over labeled biosensors. However, apart from the analyte-receptor interaction at the electrode interface, any binding resulting from incomplete separation of the sample can also give an output signal which is difficult to distinguish from the original signal [101]. To minimize measurement errors, sample preparation and sensor design studies should be carried out meticulously.

The capacitive transducers provide many advantages in biosensors, such as high sensitivity, low operation power, low loading effect because of high input impedance, and good frequency response. Capacitive transducers in biosensing have been demonstrated to be very effective in a variety of applications, including cancer tracking [102], bacteria growth monitoring [103], chemical solvent detection, DNA hybridization [104], and virus detection [105]. With capacitive transducers, biosensors can detect the presence of a wide spectrum of substances, regardless of the variety of the target molecules (analytes) in contact.

Capacitive transducers have been widely investigated for potential integration into smart wearables for human health monitoring. Just as in the affinity-based capacitive biosensors, capacitive transducing also forms the basis of flexible pressure and strain sensor systems yielding electronic skin technology (smart skin), that aims to represent the next generation of biosensing applications such as monitoring the specific motions of robotic arms and prosthesis, and human–machine interface [106]. Smart skins have been inspired by the ability of human skin to convert external pressure and strain signals into electrical signals. According to the pressure sensing properties of biological skin, pressure sensing can mainly be used for low-pressure range (0–10 kPa) pulse and micro-touch applications, while such capacitive pressure sensors can be used for surface pressure distribution applications in the high-pressure range (10–100 kPa). As the applied pressure rises to a few kPa, the sensitivity decreases significantly, which, in turn, limits its practical applications. Due to the distance between the arterial trees and the skin surface, the amplitude of the pulse-sourced mechanical waves is lost in the propagation path through the soft tissue, yielding a weak mechanical on our skin surface. Therefore, the pressure range and sensitivity of the sensor are crucial for the implementation of smart skin applications [107]. In recent years, significant progress has been made in improving the sensitivity of capacitive pressure sensors to detect ultra-low-pressure changes [108]. In addition, it is pivotal to examine as many reversible pressure changes as possible and to determine whether the capacitance change is stable or not [109].

On the way to develop electronic skins, elastomers (e.g., polydimethylsiloxane, silk fibroin, polyurethane, polyethylene terephthalate, etc.) are used as capacitive transducer materials acting as flexible matrices to deform the conductive filler network (e.g., carbon materials, metal nanoparticles, liquid metals, and conductive polymers) that cause changes in capacitance, conductivity, or resistance [110]. They convert external pressure and strain into electronic output to measure bio-parameters or the interaction of a body part to an external stimulus such as pressure. The performance of such sensors depends on the properties of the matrix, the mesh of various materials, and the interactions between them. Compared to conventional sensors based on rigid semiconductors, metals, and ceramics, elastomers are advantageous since they exhibit the highest level of strain behavior for wearable applications. Polydimethylsiloxane (PDMS) microstructures have been used to increase the sensitivity of pressure sensors; however, the sensitivety value was found to be less than 1 kPa^−1^ even in the low-pressure range [111]. On the other hand, it was stated by Mannsfeld et al. [112] that the fabrication of microstructured-PDMS films requires a challenging 4-step process. As an alternative solution to the low-pressure sensitivity drawback, it was proposed to integrate organic thin-film transistors for diversification of the sensing mechanism. With the inclusion of micro-structured PDMS as a dielectric layer in a flexible organic thin-film transistor (OTFT)-based pressure sensor, a high sensitivity that is enough (8.2 kPa^−1^) to allow the use of flexible pressure sensors in mobile health monitoring was achieved for cardiovascular medicine [113]. However, this high sensitivity was only possible in the very narrow pressure range due to the use of PDMS for the dielectric layer. In another report, the sensitivity of the pressure sensor formed by the combination of PDMS elastomer, polymethyl methylacrylamide (PMMA), and silver nanowire was calculated as 3.8 kPa^−1^ [114]. Alternately, the pressure sensitivity of a capacitive sensor prepared with the carbon/silicon structure was found to be in the very high range (0–700 kPa) but it was found to exhibit very low sensitivity (0.025 kPa^−1^) [115]. Results of the studies using capacitive transducing for pressure-sensing capabilities [108,112,113,114,115,116,117,118,119,120,121,122] are interpreted quantitatively and presented in Table 1 for easy comparison. According to the reported results, high filler content is often required to form a conductive network in the elastomer matrix to achieve measurable electrical signal amplitudes [106]. In contrast, such conductive composite materials show low breakdown stress limiting the design parameters in wearable applications.

### 2.2. Optical Biosensors

The most predominant type of biosensors is optically transducing biosensors (optical biosensors) owing to a variety of the optical methodologies to transduce the biologically generated signals. Optical biosensors are widely used and inestimable tools for medical research, including clinical diagnosis of genetic diseases, drug design, neuroscience, healthcare monitoring, protein detection, and identification [123]. Optical detection is based on the interaction of the optical field with bio-receptors where the analytes bind and trigger a biochemical reaction [124]. Optical biosensors can be categorized as “label-free”, and “label-based” depending on the targeted binding or simultaneous detection of a range of biomarkers. Commonly reported optical biosensing techniques are based on spectrometry, fiber-optics (FOBs) [125], interferometry [126], and surface plasmon resonance (SPR) [127,128]. Moreover, similar to the conventional optical phenomena, absorption, fluorescence [129], refraction, optical diffraction [85], phosphorescence, and Raman scattering are highly incorporated in optical biosensing. With the commonly used optical methods, one can measure the spatial and temporal properties of the light signal resulting from a biological event, such as the modulation of amplitude, decay time, polarization, phase, and/or energy that provide diverse knowledge about the properties of an analyte or event [130].

Optical biosensors empower the user over case-specific application areas with the advantages of real-time detection, continuous interaction at a low-cost [127,128], compatibility with small volume samples, remote sensing in out-of-reach areas [123,124,125], and fast detection [131,132,133,134]. For instance, FOBs can be adapted for the detection of a variety of bio-events from the growth of Escherichia coli to the ovary cells of hamsters. In addition, the SPR technique makes the identification of molecular binding possible by offering high sensitivity and label-free detection [128]. SPR imaging is also used in clinical studies for screening biomarkers and therapeutic targets [127]. On the other hand, evanescent-wave biosensors reflect great detection sensitivities in a short period of time and are capable of disease diagnosis with high accuracy due to their ability to assess kinetics and affinity of interactions by optically monitoring biomolecular interactions concurrently [135,136]. Immunochromatographic test scripts and lateral flow immunoassays have been present for a long time in the field of health monitoring, and such scripts and immunoassays have a wide range of use from home pregnancy test kits to the recently developed SARS-CoV-2 diagnostic kits [136].

In the wearable field, the integration of optical biosensors into compact electronics yielded wearable optoelectronics for the continuous and noninvasive extraction of vital signs. Advancements in optoelectronic integration provided wearable photoplethysmography (PPG), in which a light source sends light to the skin, and modulations of reflected/transmitted light are measured to record heart rate and related cardiac parameters [34]. The wearable devices that contain PPG have shown an abrupt increase recently [137], and today, almost all fitness trackers commonly make use of this technology together with electrocardiography (ECG) sensors to provide the most accurate results by comparison and compensation of both measurement technologies. In PPG, the vessels absorb the incident light at a known wavelength (commonly at visible and near-IR spectra) and the reflected light reaches the light sensor (photodetector, PD) where the pulse modulated light intensity is registered as heart rate (HR). Unlike ECG, PPG depends on the local profusion to extract the HR yielding freedom of location for the measurement sites on the body. However, the thickness and structure of the skin are important factors in vital sign extraction. Skin conditions may change depending on personal parameters such as age, sex, or due to a medical condition [138]. In principle, human skin provides a unique interface for wearable devices to extract physiological parameters, however, in a large number of wearable transducing mechanisms (e.g., electrochemical biosensors) the information received with the help of biosensors is removed from the surface of the skin [138]. For instance, conventional ECG electrodes that are attached to the skin surface sense the electrical signal sourced from the heart, therefore, the HR measurement can deviate up to a skin condition factor such as body fat [139]. On the other hand, PPG uses a light source of a certain wavelength (400–1000 nm) that can provide a maximum penetration reaching the dermis and hypodermis, hence, it can bypass the anatomical factors that interfere with the measurements [34]. In parallel, the effect of skin tone has also been discussed frequently in the context of optical heart rate wearables [140]. Based on the classical skin phototype classification of Fitzpatrick [141], previous experimental studies revealed that the PPG HR measurements vary up to 15% due to the melanin density difference between dark and light skins [140]. Therefore, the effect of skin tone should be considered a major factor in the construction of optical transducers for biosensors and wearables.

The PPG signal contains vital information that can be extracted as respiratory rate (RR) [142], blood oxygen saturation (SpO_2_) [143] (Figure 4A), blood pressure (BP) [144], and cardiac output [145]. Moreover, it was demonstrated to be possible to extract clinical physiological parameters directly from the measured PPG signal such as vascular assessment and autonomic function [146].

The clinical and technological research on PPG has reached its top speed with the integration into wearables and consumer electronics. To that end, breakthrough skin conformable pulse oximeters using material technologies have been reported. Yokota et al. have demonstrated an ultra-flexible pulse oximeter (Figure 4B) wrapped on a finger [147]. Unlike the rigid conventional oximeters, the authors have used flexible polymer light-emitting diodes (PLEDs) and organic photodetectors (OPDs) that can be laminated on the skin. By integrating green and red PLEDs together with an OPD, the authors have created a fully flexible pulse oximeter. The authors performed a PPG with an ultra-flexible pulse oximeter at two different wavelengths to find the blood oxygen saturation by differentiating the absorbance percentage of oxygenated and deoxygenated hemoglobin. The reported device can register a SpO_2_ change of 9% (between the oxygenation states of 99% and 90%) (Figure 4C,D). The authors have reported the stable device operation at ambient conditions for the long term and the ultra-flexible device structure can withstand a fairly good amount of applied mechanical stress (300 cycles of stretching for OPDs and 1000 for OPDs bending down to 100 µm).

Similarly, Kim et al. have demonstrated a wireless epidermal oximeter that can be laminated on human skin [148]. The oximeter includes a red (625 nm, InGaAIP) and commercially available infrared (IR) LED, a photodiode, and the complementary circuitry that are strain engineered to the stretchable configuration (Figure 4E,F). To achieve the required device stability in the optical detection of SpO_2_, Kim et al. have electrically and physically insulated the metal traces and located them on the neutral plane of the elastomeric substrates [148]. The authors have implemented the near field communication (NFC) technology to wirelessly measure blood pulse oxygenation and used an ultrathin medical adhesive to bind the devices to the skin. To block the environmental light, the demonstrated epidermal oximeter uses a black textile and an astable oscillator controls the current in the LEDs [148]. This way, the authors detected the variations in the concentration of oxyhemoglobin (ΔO_2_Hb) and deoxyhemoglobin (ΔHHb) by the photodetectors’ responses during red and IR illuminations. The demonstrated wireless oximeter devices were simultaneously operated with a commercial NIRS (Near IR spectroscopy) oximeter and variations in the ΔO_2_Hb and ΔHHb were recorded from the adjacent regions of the forearm (Figure 4G,H). A good agreement in the oxygenation results, mechanically flexible form factor, and wireless data and power communication promise for future use of the skin conformable devices as a gold standard in the wearable field.

Another commonly used wearable optical transducing method is colorimetry. Colorimetric transducers have the key enabling properties to develop wearables, such as easy detection and user-friendly feedback by the simple color-changing feedback mechanism. Colorimetric wearables have been commonly demonstrated to be laminated on the skin surface and give the user necessary optical feedback by the color change of the active transducers [149,150]. Choe et al. have demonstrated a wearable colorimetric patch based on a thermoresponsive plasmonic microgel embedded in a stretchable hydrogel film [151]. The authors have demonstrated plasmonic microgel film that undergoes large and reversible color shifts with respect to temperature changes without the change in the overall sensor volume (Figure 4I). To control the colorimetric response, the authors fabricated raspberry-shaped plasmonic microgels by decorating them with gold nanoparticles (AuNPs) on thermoresponsive poly(*N*-isopropylacrylamide) (PNIPAM) hydrogels. To provide the repetitive colorimetric response under the successive heating/cooling cycles of the colloidal solution, the authors have incorporated the plasmonic microgels into the flexible polyacrylamide (PAAm) hydrogel film and realized reliable thermoresponsive color shifts. The thermoresponsive plasmonic structure is encapsulated in two PDMS (polydimethylsiloxane) films with thicknesses of 150 µm to form the mechanically flexible device structure to operate on human skin (Figure 4J). This way the wearable colorimetric patches undergo an efficient peak shift of 176 nm (545 nm to 721 nm), leading to a high-contrast colorimetric response, and devices are reported to exhibit a stable operation after 10 cycles of heating and cooling (Figure 4K,L).

**Figure 4 biosensors-12-00385-f004:**
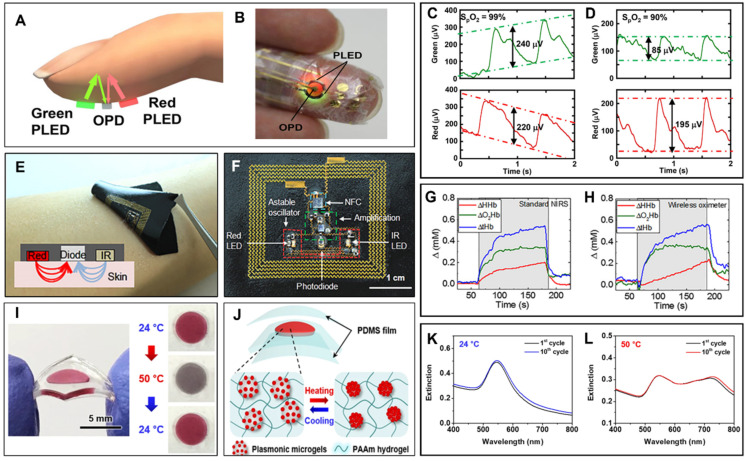
Wearables using optical transducers. (**A**) Schematic illustration of the reflective pulse oximetry from finger [147]. OPD records the reflected light intensity that is illuminated synchronously by the green and red PLEDs to extract the PPG and the resulting SpO_2_. (**B**) Photograph of Yokota et al.’s ultra-flexible polymer oximeter wrapped on the finger [147]. (**C**) Demonstrated SpO_2_ extraction at green and red wavelengths by the change of the PPG signal intensity [147]. (**D**) A 9% change in the SpO_2_ value (from 99% to 90%) yields a detectable intensity change in the resulting PPG signal at both wavelengths [147]. (**E**) Epidermal wireless pulse oximeter demonstrated by Kim et al. [148]. The device is skin conformable by the medical adhesive and the strain engineered device structure. (**F**) Components of the wireless epidermal oximeter [148]. The device includes NFC technology to wirelessly detect PPG and SpO_2_. All device components are encapsulated and located on the neutral plane of the elastomeric substrate, yielding the mechanically robust device operation on the human skin. (**G**,**H**) Demonstrated simultaneous measurements from the commercially available NIRS bulk oximeter and the epidermal wireless oximeter [148]. Although the two systems use different configurations to optically extract the hemoglobin concentration, the resulting curves exhibit similar trends and values. (**I**) Choe et al.’s colorimetric thermoresponsive wearable [151]. The incorporated plasmonic microgel structure undergoes an efficient spectral shift during the heating and cooling processes, yielding colorimetric feedback to the user about the skin temperature. (**J**) The demonstrated plasmonic microgels are embedded in PDMS films to provide a device structure with enhanced mechanical stability [151]. (**K**,**L**) Recorded spectral shifts during 10 cycles of heating/cooling [151]. The wearable colorimetric patches exhibit a reversible color change with a 176 nm peak shift from 545 nm to 721 nm yielding high-contrast colorimetric feedback to the user. (**A**–**D**) Reproduced with permission from [147] under the terms of the Creative Commons Attribution-NonCommercial license, Copyright © 2016, American Association for the Advancement of Science. (**E**–**H**) Reproduced with permission from [148] under the terms of the Creative Commons Attribution-Noncommercial license, Copyright © 2016, American Association for the Advancement of Science. (**I**–**L**) Reproduced with permission from [151] under the terms of Creative Commons Attribution 4.0 International License, Copyright © 2018 Springer Nature.

### 2.3. Thermal/Calorimetric/Thermometric Biosensors

Biosensors using thermal/calorimetric/thermometric transducers measure energy changes (heat, *q*) of a system and its surroundings. Such biosensors are constructed by immobilization of bio-elements onto temperature sensors, in which the bio-recognition sourced energy change of a system and its surroundings (i.e., heat exchange) is determined [152]. The working principle of such biosensors involves: (i) the entrance of analyte solution through the substrate inlet and the measurement of temperature by a thermal transducer (e.g., a thermistor [153,154] microelectromechanical system, thermocouple, resonator, or thermophile [155]), (ii) the flow of solution through a packed column consisting of immobilized enzymes where enzyme-catalyzed reactions occur, resulting in the loss or generation of heat, and (iii) the remeasurement of temperature by a separate thermal transducer while solution proceeds towards outlet [156]. This way, the thermal transducers quantify the change of heat that occurs within endothermic and exothermic enzyme-catalyzed chemical reactions in biological systems to interpret findings with respect to the analyte concentration, molar enthalpy, and product formation or the total number of molecules in such reactions. To detect the temperature change from first principles, we consider the total heat in the system as:*q* = −*n*_p_ (Δ*H*)
and,
*q* = −*C_p_* (Δ*T*),
yielding a temperature change:Δ*T* = −(Δ*H*) *n*_p_/*C_p_*
where *q* is the total heat, *n*_p_ is the number of moles of the product, Δ*T* is the change in temperature recorded by the enzyme thermistor (ET), and Δ*H* is the molar enthalpy change. *C_p_* is the heat capacity of the system including the solvent [157].

Thermal transducing is also commonly used in resistive devices to reflect the temperature change as a change of resistance. The change in temperature with respect to resistance can be described by the Steinhart–Hart equation:1/*T* = *A* + *B*(ln *R*) + *C*(ln *R*)^3^
where, *A*, *B*, and *C* are the experimentally derived coefficients. From this, the relation between the resistance at temperature *T*, ($R_{T}$) and the resistance at *T* = 0, ($R_{T_{0}}$) can be written as:$$R_{T}=R_{T_{0}}e^{\beta (\frac{1}{T}-\frac{1}{T_{0}})}$$
for narrow temperature ranges where *β* is a material constant (ranging between 4000 and 5000 K) [158].

Starting with the conventional ET devices containing immobilized enzyme columns, this class of biosensor has been progressed into the micro and multi-sensing (hybrid) versions of such devices together with the commercialization [159,160,161]. Thermal transducers are highly suitable for a wide range of applications in the detection of bioprocesses with a groundbreaking development of thermometric enzyme-linked immunosorbent assay (TELISA) combining the fundamentals of ELISA [154,162] for the determination of hormones, antibodies, and other biomolecules generated during the fermentation process, environmental monitoring [163], clinical diagnosis [164], and food analysis [165].

Based on the abovementioned literature, the current progress promises the developments aiming to design microdevices with multi-analytical capabilities in addition to portability and digital data reporting. For instance, periodic and conventional measurement of body temperature with a thermometer can help regarding early diagnosis of malfunctioning metabolic processes of the body and symptoms of illnesses such as fever, infection, depression, or even insomnia problems. Real-time and continuous monitoring of such measurements is accurately possible with wearable temperature sensing systems by gathering diagnostic information and understanding signals of underlying diseases. Lightweight and flexible temperature sensing devices can be attached to the human skin without the awareness of the user and provide continuous monitoring. In the market of wearables, Yono’s earbud [166], Ava Science’s wristband [167], Empatica’s watch [168], Oura’s ring [169], and VitalConnect’s chest patch [166] are great examples of functioning as a thermometer, temperature sensors, a skin temperature measurand, and a thermo-resistor, respectively.

### 2.4. Gravimetric/Piezoelectric/Mass-Sensitive Biosensors

Piezoelectric biosensors are formed by the coupling of a bio-component with piezoelectric transducers that are usually based on a quartz crystal coated with gold electrodes [170]. Piezoelectricity is a reversible process in which an electrical charge accumulation is induced due to the applied mechanical stress causing deformation or vibration of the material [171]. Piezoelectric biosensors work upon the detection of frequency changes occurring on the transducer surface. When the target analyte attaches to the material, the resulting mass shift on the crystal component causes resonance frequency alterations. To that end, physical, chemical, or biological microcantilevers detecting changes in cantilever bending or vibrational frequency can be categorized as gravimetric/mass-sensitive biosensors.

In the construction of piezoelectric biosensors, the end application plays a highly important role, since the piezoelectric materials and designs differ for specific purposes such as label-free detecting [172] bacteria or virus [173], or cancer biomarkers [174]. Therefore, the selection of the mass-sensitive transducer elements includes a variety of factors ranging from the type and the thickness of electrodes (e.g., gold, chromium, platinum, titanium, etc.) to the detected bio-agents (e.g., warfare agent, virus, bacteria, etc.). Owing to various construction options and possible features, there is a high demand for piezoelectric transducers and their biosensing applications. While the precision and sensitivity are the advantageous specifications, the common downside of mass-sensitive transducers is the temperature-dependent sensitivity yielding the loss of such features at extremely low or high temperatures. In that sense, any change in the temperature causes thermal inconsistency of various features (e.g., dielectric, piezoelectric, or electromechanical) and weakening of acoustic waves and dielectric losses. All these factors should be considered while constructing a piezoelectric biosensor to maintain the effectiveness of the sensor at various temperatures [175].

With the demonstration of in situ interfacial mass detection by piezoelectric transducers [176], piezoelectric/gravimetric/mass-sensitive biosensors have been widely used in the medical field as immunosensors for the detection of bacteria and viruses. Moreover, piezoelectric genosensors for the detection of DNA or RNA fragments regarding their distinct sequence of bases have been reported [177]. Important applications of this class of transducers include cell and tissue characterization, healthcare monitoring, pressure sensing, and detection of endotoxins [178,179] cholesterol [180,181], pesticides [182], and breast cancer [183,184]. Moreover, piezoelectric biosensors show potential for applications in food quality detection, and environmental and clinical analysis [185].

Piezoelectric transducers have found a wide spectrum of wearable device applications. Han et al. have demonstrated a self-powered electronic skin based on a piezo-biosensing [23]. The authors have implemented enzyme/ZnO nanoarrays as the basis of the piezo-biosensing unit and demonstrated an electronic skin that can detect lactate, glucose, uric acid, and urea in the perspiration [185]. Very recently, Su et al. have demonstrated muscle fibers inspired piezoelectric textiles for wearable physiological monitoring [186]. To mimic the muscle fibers, the authors have used the surface modification of polydopamine (PDA) and monitored the real-time heart rate and dynamic output profile for the voice recognition with the skin attachable wearable. Alternately, piezoelectric transducers are frequently used as energy harvesters in stretchable and self-powered wearables and implantable devices [187]. Dagdeviren et al. have reported biocompatible piezoelectric energy harvesters based on the piezoelectricity of ZnO material [188]. Similarly, Zhu et al. have reported piezoelectric nanogenerators that are mechanically flexible for wearable applications [189]. With its key properties, nanomaterial-based piezoelectricity presents a compatible platform as transducers for wearable biosensing and energy harvesting. This multi-discipline approach may lead to a new developing orientation of health and fitness wearables by promoting such flexible self-powered multifunctional nano-systems.

## 3. Supplementary Technologies for Wearable Biosensing

In this section, we provide a general outlook on the supplementary technologies empowering the wearable use of transducers for biosensing. We summarize and highlight the enabling supplementary technologies for wearable transducers to represent a user-friendly and low-cost platform by providing key factors, such as ease of access, continuous and non-invasive analysis, mechanical robustness, enhanced user adoption, and usability. Furthermore, we extend our perspective to the energy sources, the data communication technologies, the location and position services, and the biocompatibility within the supplementary technologies providing a framework for the development of wearable biosensors.

### 3.1. Microfluidics and Biomedical Microelectromechanical Systems (Bio-MEMS)

Microfluidics provides an efficient platform for transducers by forming the core of the sensing elements and providing advanced liquid holding and storage capabilities [190]. Microfluidics-based point of care medical sensors are of ever-growing interest due to their versatility as wearable lab-on-the-body systems [191,192,193]. This section is dedicated to a brief outlook on the importance and varieties of the conformal biomedical microelectromechanical (bio-MEMS) sensors as a supplementary technology to the wearable transducer-based sweat analysis. Sweat is a compatible substance with wearable medical sensor systems owing to its transparent and hypotonic properties [191,194]. Furthermore, natural sweating induces high enough pressure (~70 kPa) to sustain pressure different for the microfluidic channels [191,194,195]. However, sweat-based sensing is prone to false-concentration-based analyte readings due to vaporization [191,194,196,197] and inhomogeneous local sweat gland density [195]. It has been demonstrated that the measurement error could be as high as 114% [192]. Furthermore, the sensory system should include microfluidics, electronics, and a power supply to collate and relay data on the collected and analyzed sweat in real-time, which results in a rather bulky system. Moreover, the said system should be conformal if desired to be employed in direct contact with the skin on different parts of the body. Despite these shortcomings and constraints, analysis of the constituents of sweat is a crucial part of medical diagnosis with an ever-growing focus owing to the variety of critical biomarkers that can be detected in tiny volumes.

Biomarkers present in the sweat that can be detected by wearable microfluidics-based medical sensors are (i) glucose, (ii) lactate, (iii) pH, (iv) chloride, (v) creatinine, (vi) tyrosine, (vii) uric acid, (viii) potassium, (ix) sodium, (x) ascorbic acid, (xi) cortisol, (xii) dopamine, and (xiii) adrenaline. The concentration of glucose in sweat is 1% of that of the plasma and its accurate measurement is very important for continuous monitoring of diabetes [193,198]. On the other hand, lactate is an indicator of physical effort signaling the transition from aerobic to anaerobic metabolism [194,199]. Determining the level of pH helps in ascertaining the neuromuscular condition of the subject [194], while chloride levels are used to help with the diagnosis of cystic fibrosis [194,195]. The level of creatinine is an important indicator in confirming the hydration status of the metabolism and renal health [195] and determining tyrosine level from sweat is generally used for the diagnosis of liver diseases and various psychiatric disorders [197]. Thus, the analytes found in sweat serve as crucial markers for the performance of vital organs and critical systems in the human body; therefore, using sweat as a working fluid in a wearable Bio-MEMS sensor is crucial. The reported wearable microfluidic devices that use sweat to analyze the abovementioned analytes are summarized in Table 2.

In the wearable microfluidic wearable Bio-MEMS, the sweat is collected and regulated via the natural pressure difference of sweat glands [191,192,194,195,196,197,198,199,200,201,202]. The capillary force is usually strong enough to drive the fluid through microchannels leading to the chamber where various sensor types are employed [191,195,198]. The flow regulation can be managed via active thermo-responsive hydrogel valves controlled by microheaters [192], or passive capillary pressure-bursting valves [200]. Capillary force arises due to the hydrophilicity of the wetted surface and allows liquids to travel long distances without large pressure difference when the characteristic length-scale of the ducts is in microns [195,198]. The microchannel itself will exert a shear resistance on the flow, which will be balanced with the capillary force, and a static pressure build-up will occur when there is no flow. This pressure might be released in terms of sudden hydrodynamic pressure by changing the shape, and thus the contact angle, along the way. A bursting valve operates on the very same principle [200]. On the other hand, an active valve, such as the thermo-responsive system [192], does not require plugging the duct entirely but rather partially obstructs the duct, inducing substantial head loss on the flow thus depriving it of the mechanical energy necessary to overcome the shear resistance. The reported works using these main driving mechanisms for the control of sweat flow in wearable Bio-MEMS are summarized in Table 2.

#### 3.1.1. Energy Sources and Detection Mechanisms

Although passive effects such as the capillary force or secondary ducts do not require a power source, active valves, as well as sensory electronics along with powering the detection mechanism and achieving real-time or ad-hoc relay of data, require a power supply. Power supply in wearable Bio-MEMS sensors is based on either energy scavenging or electrochemical conversion aside from the fact that microfluidics relies on natural pressure difference of sweat [194,195] except for thermo-responsive valves that are polymers actuated via a change in temperature [200]. The power source exploited in such systems should be of low voltage and current for the safety and practicality of the system. Also, the source should be either easily replicable or rechargeable, avoiding the requirement for complicated reassembly procedures. To that end, the most practical common solution is to employ batteries to generate electric currents through electrochemical conversion through battery packs. On the other hand, the electronics can be fitted with energy scavenging MEMS hardware using electromagnetic fields of a certain frequency, such as the one emitted by a smartphone [200], or the body movement of the subject wearing the system transforming mechanical vibration to electricity by means of induction and triboelectric effect during physical exercise [201]. Furthermore, the external energy might only be required for near field communication (NFC) to pair the wearable Bio-MEMS sensor and the device employed to collect the data [195]. Furthermore, smartphones can incorporate the wearable system in a vast IoT framework [191]. The reported microfluidic wearables using various energy sources and the utilized detection mechanisms are summarized in Table 3.

#### 3.1.2. Data Transmission

Another important aspect for wearables to represent a mobile technology is their data transmission. To that end, a wearable system would usually exploit wireless means of communication. The wireless connection usually suffers from a limited onboard power source with wearable systems; therefore, long rage real-time radio frequency (RF) communication is not feasible, whereas short-range data transfer options are widely preferred. There are currently three main methods in the data communication of wearables, namely, wireless, i.e., Bluetooth [192,194,196,197,201], NFC [195,200], and visual data capture [191,195,200]. Bluetooth communication can be real-time while NFC and visual data recording are designed to be intermittent. While visual data recording does not require either special electronics or power sources dedicated to the task, wireless and NFC need custom circuitry, an RF antenna, and a device to be paired and collate the data of interest [192,194,195,196,197,200,201]. In addition, active flow control can be achieved by wireless communication via a mobile device [192]. The presence of integrated electronics increases the device in size to a relative bulk, however, still practical to apply on the skin.

The transmitted sensory data are of different origins. There are simplistic systems using NFC to pair the sensor with a device such as a smartphone to start an application for image processing out of colorimetric data [195]. Colorimetric data is obtained by the reaction of analytes with certain chemicals and enzymes that emit certain frequencies if excited by fluorescent or visible light [191,195,200] that will reveal the spatial concentration. These systems do not necessarily require real-time data transfer and additional onboard power dedicated to the colorimetric analysis [195]. In addition to colorimetric sensors, there are systems with hybrid sensing compartments or completely different transducing approaches, e.g., galvanic, strain, piezoresistive effect, and electrochemical. The reported devices with various transducing mechanisms are summarized in Table 3.

Electrochemical biosensors are exclusively microfluidic and rely on custom electrodes coated with specific mediator molecules, such as carbon nanomaterials, to choose and react to a targeted analyte [196,197,199]. The induced current by the transducer is directly related to the flux of the molecules of interest in the vicinity of the electrode [192]. Thus, the transportation and diffusion of analytes in the microfluidic network and reservoirs are of utmost importance [197]. Also, strain sensors could exploit swelling of bulk material, e.g., hydrogels, as sweat gets absorbed while stretching a conductive fabric changing its resistance [202]. Similarly, the sweat rate can be detected based on the change in galvanic properties of specifically designed microchannel geometries.

#### 3.1.3. Biocompatibility

In microfluidic wearables, the sensors and antennas, along with a printed electrical interface to the necessary PCB, are expected to be implemented on biocompatible conformal structures to seamlessly fit on any surface. Furthermore, the sensor is supposed to be attached to the skin via medical-grade adhesives. Sweat itself is supposed to be collected and stored by the carefully tailored network of microfluidic channels and chambers embedded in this elastic structure. Such a network can be implemented in PDMS [196,198,199,201], silicone rubber [194], and medical adhesives [197].

Polyimide (PI) [197,198], PDMS [193], and polyethylene terephthalate (PET) [197] are demonstrated as a cover to the microchannels, as well as for the printing of the sensory electrodes. Moreover, different porous materials could be used to collect sweat, such as hydrogel [202] and Ecoflex [198], in certain volumes. Likewise, paper and cotton can be used for sweat collection, fluid flow compartmentalization, and sensor placement [191,193]. In addition, the thermo-responsive valves can be manufactured out of poly(*N*-isopropylacrylamide) [192].

To reach the desired wearable microfluidic device components, microchannels and reservoirs can be implemented via different fabrication techniques such as photolithography, screen printing, laser engraving, laying cotton fibers, and embossment. Table 4 presents the list of the reported wearable fabrication techniques and biocompatible microfluidic device materials. The said techniques are utilized to manufacture tailored microchannels and electrodes meeting different production constraints. For instance, laser engraving is utilized as a faster and cheaper solution as it reduces the need for infrastructure and expertise required for photolithography [195,196,197,199,201,203]. Also, making use of easy-to-obtain materials such as cotton or paper when feasible greatly simplifies the manufacturing procedure and reduces the overall cost [191,193]. Likewise, the techniques of screen-printing and embossment are employed as a cost-effective solution to microchannel manufacturing while preserving the dexterity to obtain a channel network with bifurcations [194,195,196,197]. Nevertheless, manufacturing electrodes for electrochemical or galvanic sensing or building the associated interfacial electronics entails the use of photolithography [195,196,199,203].

Overall, the end product is biocompatible and flexible so that the wearable biosensor will perform without exhibiting any chemical reaction to the skin or the sweat. To that end, it is important to acknowledge that the flow rates in these microfluidic networks are usually on the order of 0.1–1 μL/min and they are designed to accommodate fluidic conditions of natural sweating.

### 3.2. Location and Position Services

Currently, available wearable devices are mostly equipped with location/position services for various purposes such as finding a route, counting the steps, or calculating exercise output to give extended feedback to the user. Wearables are more efficient for location services than smartphones since they are bonded to the user in various form factors. This section provides a general outlook on current advancements in the location and position systems as a supplementary technology for wearables.

The global positioning system (GPS) used in outdoor positioning is a satellite-based navigation system developed by the United States Department of Defense [204]. Outdoor position information can be calculated when four or more GPS satellites are in the line of sight [204]. The GPS enables critical outdoor positioning, predominantly for civil, commercial, and military applications. To that extent, Assisted-GPS (A-GPS) technology is a positioning system used in mobile devices to find the user’s location within the A-GPS address server via the base station [205]. The accuracy provided by GPS in outdoor positioning is between 3 m and 15 m, while usually around 10 m for indoor positioning [204,205]. On the other hand, the accuracy of the A-GPS in outdoor positioning is 15 m while limited to 50 m for indoors [205]. Although these accuracy values allow sufficient position determination outdoors, since people spend more than 80% of their time indoors, suitable indoor geolocation systems are in high demand. Unfortunately, the use of GPS satellites in indoor localization is not sufficient enough with the weakening and loss of GPS signals due to the atmospheric delays, signal reflections (multipath), steel structures, roofs, and building walls [204,205]. Therefore, in the last two decades, a serious amount of work has been carried out on developing new technologies that enable a reliable indoor positioning with high accuracy and low average error. When the GPS signals cannot be reached, the positioning problem is solved by using different technologies such as infrared, ultrasonic, cellular, radio frequency recognition (RFID), wireless network (Wi-Fi), Bluetooth beacon, or ultra-wideband (UWB) sensors [206,207,208,209]. In some studies, even visible light [210,211] and technologies that use the earth’s magnetic field have been used [212,213]. However, new algorithms and methods are needed to improve the results.

Among the new indoor positioning methodologies, ultra-wideband (UWB) sensor technology steps forward with its ideal indoor distance estimation, indoor geolocation, indoor tracking, and navigation [214]. The UWB sensor technology is important in providing centimeter-scale positioning accuracy indoors. UWB is a radio technology used in short-range high-bandwidth communication. UWB has higher bandwidth of 500 MHz; therefore, signals usually reach the receiver in more than one way [215]. However, high bandwidth allows a range of frequencies to be used at different times; thus, UWB can be used to solve multipath problems and interference effects. UWB transmitters consume relatively low power compared to other indoor geolocation technologies, making them a more efficient option by providing a longer battery life for wearables. The power consumption of wearable UWB transmitters is generally less than 1 mW, while the power consumption of UWB receivers is around 400 mW [216].

The UWB frequency range for communication applications is between 3.1 and 10.6 GHz [217]. This frequency range makes UWB signals less affected by disturbances and prevents them from being affected by Wi-Fi and Bluetooth signals, mainly operating in the 2.4 GHz frequency band. In indoor positioning, where the number of people and objects is high, the field of view (LOS-line of sight) may be relatively blocked, which may cause delay and deviation in the received signal. It is important to note that the error rate increases if the user to be positioned is out of sight, but it is difficult to conclude that the absorbing effects of the human body will increase these errors [218,219]. Time of flight (ToF) and time of arrival (ToA) methods are used to determine indoor location with UWB sensors.

UWB sensors with the time difference of arrival method (TDoA) draw attention in medical applications [220,221]. Another technique, the angle of arrival (AoA), allows the signals to be compared with the signal strength of the angles received from at least two sources. In this way, the object’s location can be found from the angle of the intersection of the signals [222]. With the ToF and TDoA methods, indoor positioning can be performed with an average error of around 20 cm [223]. Since UWB technology requires a special transmitter and receiver infrastructure, it has not yet entered the informatics market, except for a few industrial applications [224]. However, the current advancements and planning for the integration of UWB sensors in mobile phones show that UWB sensors will be used in a wide variety of areas in the very near future, including wearables [225,226].

## 4. Conclusions, Discussion, and Outlook

In this review, we have provided a detailed outlook on the transducers for biosensors and their wearable application, including their construction, classification, and wearable uses. We have also summarized the recent progress in wearable biosensors and the role of transducers in the development of enabling applications ranging from personal health and fitness tracking to clinical and environmental uses.

Transducers and their integration into biosensors and wearables have paved the way toward a radical shift in the main trends of consumer electronics and the application of western medicine. To that end, the development of wearable biosensor-based telehealth systems may provide real-time control of personal health parameters without physically visiting a healthcare unit.

Design and implementation of new material-based technologies allow transducers that are not possible by the conventional methodologies. Together with the recent progress in supplementary technologies, such as microfluidics, wireless data communication, and location/position services, wearable biosensors present viable solutions in a diverse application spectrum from healthcare-focused devices to enzyme electrodes, providing the transformation of glucose into a detectable current output via oxygen reduction, bn which they prepare, organic [137], and piezoelectric polymers [186] as active transducers and the formation of electronic components on unconventional substrates such as paper [227], textile [228] and silk [229], together with strain engineering of device components [148], hold utmost importance for the realization of the wearable technologies with enhanced feel and form factors. To that end, we have provided a material- and design-based perspective on the transducers to be used in biosensors and their wearable applications. On the other hand, Si microprocessor technologies are not expected to be outperformed by a nanomaterial soon; however, current proof of concept demonstrations on two-dimensional material technologies provided computational opportunities with high performance and mechanical flexibility [230].

In line with the current progress of biosensors, future wearable applications are expected to monitor multiple vital biomarkers continuously and non-invasively. Clinical-grade biomarker extraction is not mandatory in the context of consumer wearables; however, the implementation of new materials and designs hold great promise for telehealth through the incorporation of wearables in clinical studies. Forming the basis of wearable technologies, transducers for biosensing may serve as a key factor to overcome the current drawbacks of public health strategies. To that end, we believe that our review and perspectives on the biosensors with highlighted breakthrough works provide a complete framework and guide the readers in the construction of biosensing devices and their wearable applications.

## Figures and Tables

**Figure 1 biosensors-12-00385-f001:**
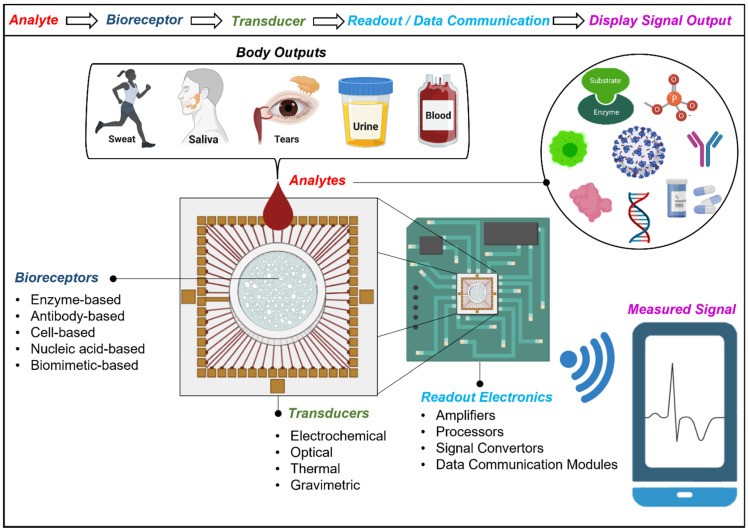
Schematic illustration of biosensing. Analyte containing bodily fluids match with the bio-receptor which can be an enzyme, antibody, cell, nucleic acid, or biomimetic-based. The matching of the analyte and bio-receptor creates a change in the signal that is registered and converted to a measurable output by the transducers. Signal conversion can be in the means of electrochemical, optical, thermal, or gravimetric. The output signal is processed by integrated or discrete electronics where the signal can be amplified, filtered, or sent to desired device platforms. For extended functionality and mobility, modern biosensors provide wireless data communication to smart devices with their integrated data communication module. This way, the extracted signal can be displayed or recorded on any mobile device for personal health monitoring. (Created with BioRender.com accessed on 27 April 2022).

**Figure 2 biosensors-12-00385-f002:**
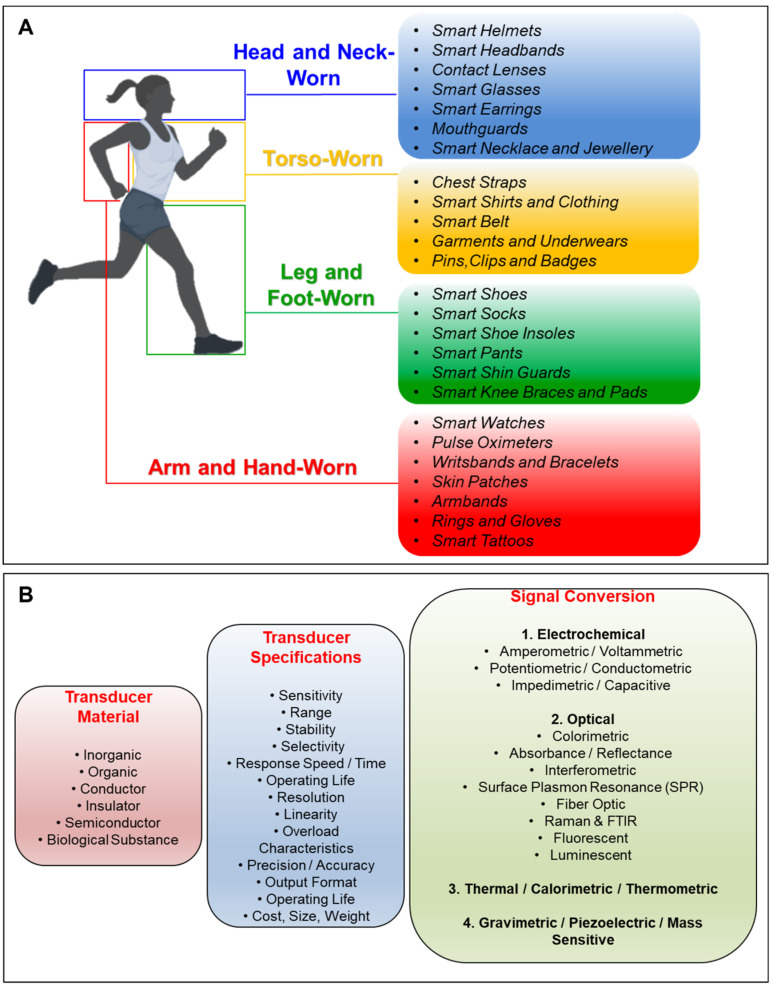
Wearable biosensors and their transducer specifications. (**A**) Body locations for which a variety of wearable form factors are reported [33,34,35]. The variety of the transducing mechanisms and the miniaturization of electronics provide wearables that can be worn on the head, neck, torso, legs, feet, arms, and hands. (**B**) Transducer specifications for construction of biosensors. Transducers are the main components in the biosensors that register the biological signal and convert it to a detectable output that can be in various formats. Active sensing material can be inorganic, organic, conductor, insulator, semiconductor, or in the form of a biological substance. While the transducer specifications are defined by the active material, the design of the transducer device can affect the resulting wearable device specifications such as sensitivity, stability, cost, and variety of the specifications that are given under the transducer specifications (middle panel of (**B**). Signal conversion mechanisms are conventionally classified as the means of electrochemical, optical, thermal, and gravimetric transducing. A more specific classification of the transducers with the signal conversion mechanisms is given in the panel on the right-hand side of (**B**). (**A**) Created with BioRender.com accessed on 27 April 2022, (**B**) Adapted from the data shown in ref. [36].

**Figure 3 biosensors-12-00385-f003:**
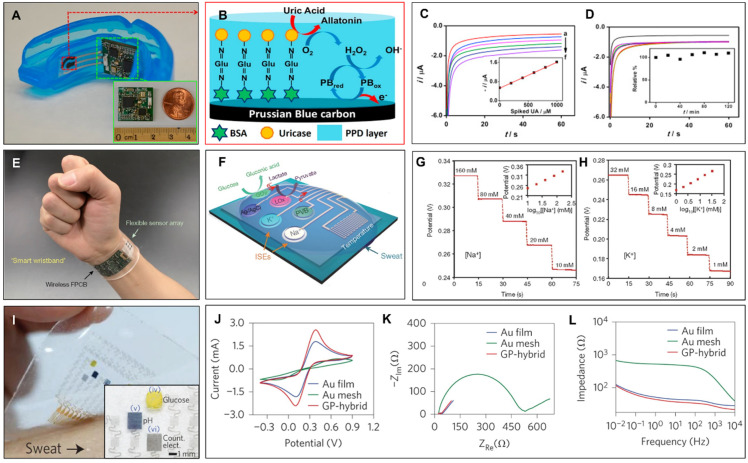
Wearables using amperometric, potentiometric, impedimetric, and voltammetric transducer mechanisms. (**A**) Kim et al.’s smart mouthguard with integrated working electrodes and PCB module with Bluetooth for wireless data communication [67]. (**B**) Schematic illustration of the reported bio-receptor yielding amperometric changes due to binding of salivary uric acid to uricase containing working electrode. (**C**) Recorded amperometric changes with the increasing concentrations steps of 0.2 mM (**A**–**F**). Inset shows the resulting calibration plot with respect to uric acid concentration. (**D**) Stability of measurements with respect to time. Authors have reported the stability of the amperometric transducers with 20 min. intervals over a total 2 h measurement. Inset shows the deviations from the original amperometric response at t = 0. (**E**) Gao et al.’s wearable-integrated sensor array allows real-time sweat analysis [68]. (**F**) Schematic representation of the array of sensors including potentiometric transducing of sodium (Na^+^) and potassium (K^+^) ions, and amperometric transducing of glucose and lactate from sweat. A resistance-based temperature sensor is also included in the array and packed together with the PCB containing a programmable microcontroller and a Bluetooth module for wireless data communication. (**G**) Experimental demonstration of potentiometric transducing of Na^+^ in NaCl and (**H**) K^+^ in KCl solutions with increasing concentration of ions. (**I**) Lee et al.’s transparent and flexible multi-sensor skin patch [69]. Demonstrated devices contain gold (Au) mesh, Au film, and Au-doped graphene as an electrochemically active layer for multiple transducing mechanisms including amperometry, (**J**) voltammetry, and (**K**,**L**) impedimetry for the detection of glucose and pH from a limited volume of sweat. (**A**–**D**) Reproduced with permission from [67], Copyright © Elsevier 2015, (**E**–**H**) Reproduced with permission from [68], Copyright © Springer Nature 2016, (**I**–**L**) Reproduced with permission from [69], Copyright © Springer Nature 2016.

**Table 1 biosensors-12-00385-t001:** Reported sensitivity ranges and pressure sensitivity values for the flexible capacitive sensors.

Material	Sensitivity Range (kPa)	Pressure Sensitivity (kPa^−1^)	Reference
Ecoflex	0–5	0.601	[108]
PDMS square pyramid microstructure	0–2	0.55	[112]
PDMS (microstructure)/PiI2T-Si	0–8	8.2	[113]
PMMA/PDMS/PVP/Silver	45–500	3.8	[114]
Carbon/Silicon	0–700	0.025	[115]
PDMS porous structure	0–0.33	0.26	[116]
ACC/PAA/Alginate	0–1	0.17	[117]
MAA/DMAPS	0–5	9	[118]
PEDOT:PSS/PDMS/silica	0–10	1	[119]
AgNW-PMMA	0–1	2.76	[120]
Graphene Micropyramid	0–4	7.68	[121]
Au/PET/PDMS micropillar	0–16	0.42	[122]

**Table 2 biosensors-12-00385-t002:** Biomarkers and driving mechanisms in wearable microfluidic Bio-MEMS *.

Analyte	Ranges	Limit of Detection	Sensitivity	Literature
Glucose	0–400 μM	1.5–7 μM	1.08–3.5 mA mM^−1^ cm^−2^	[191,192,195,198,200]
Lactate	0–100 mM	0.2–2 mM	36.2 μA μM^−1^ cm^−2^	[191,192,194,195,199]
pH	4–8.5	–	71.4 mV pH^−1^	[191,194,196,201]
Chloride	0–625 mM	5–39 mM	–	[191,195,196]
Creatinine	0–1000 μM	15.6 μM	–	[195]
Tyrosine	0–160 μM	3.6 μM	0.61 μA μM^−1^ cm^−2^	[197]
Uric Acid	0–140 μM	0.74 μM	3.50 μA μM^−1^ cm^−2^	[196]
Potassium	0.1–100 mM	–	–	[194,195]
Sodium	0.2–200 mM	–	56 mV dec^−1^	[194,196,201]
Ascorbic Acid	0.02–10 mM	0.013–10 μM	0.78 × 10^5^ C mol^−1^	[193,200]
Cortisol/Cortisol-BSA	5–100 ng/mL	–	–	[200]
	1–8 mg/mL	–	–	
Dopamine	1–100 μM	0.05–1 μM	1.1 × 10^5^ C mol^−1^	[193]
Adrenaline	10–500 μM	2–10 μM	0.8 × 10^5^ C mol^−1^	[193]
**Microfluidic Drive**	**Reported Pressure Values**			**Literature**
Pressure of Sweat Glands	70–72 kPa			[191,192,193,194,195,196,197,198,199,200,201,202]
Capillary Force (pressure difference)	100–400 Pa			[191,194,195,198]
Active Valves (thermo-responsive hydrogels)	15–300 mmHg			[192]
Passive Valves (bursting valves)	Laplace-Young Equation			[200]

* Data collated solely from the work reported exclusively in the selected references.

**Table 3 biosensors-12-00385-t003:** The reported energy sources and utilized detection mechanisms in microfluidic wearables.

Energy Source	Literature
Electrochemical Conversion (Rechargeable Battery Pack)	[192,194,196,197]
Energy Scavenging (Radio Frequency)	[200]
Energy Scavenging (Mechanical Motion)	[201]
Natural Pressure Difference (~70 kPa)	[191,192,194,195,196,197,198,199,200,201,202]
**Detection Mechanism Literature**	
Colorimetric (Fluorescent and visible light)	[191,195,200]
Strain (Swelling)	[202]
Galvanic (Capacitance)	[196]
Electrochemical (Mediator molecules)	[192,193,196,197,199]

**Table 4 biosensors-12-00385-t004:** Biocompatible materials and fabrication techniques for microfluidic-based wearable biosensors.

**Materials**	**Literature**
PDMS	[196,198,199,201]
Silicone Rubber	[194]
Medical Adhesive	[197]
Polyimide	[197,198]
Polyethylene Terephthalate	[197]
Hydrogel	[202]
Ecoflex	[198]
Paper	[191]
Cotton	[193]
Poly(*N*-isopropylacrylamide)	[192]
**Fabrication Technique**	**Literature**
Photolithography	[195,196,199,203]
Screen printing	[194]
Laser Engraving	[197,201]
Laying Cotton Fibers	[191]
Embossment	[195]

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
