# Peer review of "Transducer Technologies for Biosensors and Their Wearable Applications"

_biosensors, 2022, doi:10.3390/bios12060385_

Round 1

Reviewer 1 Report

Transducer Technologies for Biosensors and Their Wearable 2
Applications

This review, authors focused biosensors based different wearable modified electrodes. I recommend this review for publication of revision.

  1. Introduction Section: Many sentences are describing two events with are not properly linked with each other. I recommended the authors to rewrite the part to convey their message clear for the readers.
  2. There are many grammatical and typographical errors. Please check the manuscript and refine carefully.
  3. “Title: Authors must check and revise.

Reviewer 2 Report

  1. In line 419, in my opinion, there is a typo “modulating the of dielectric properties”
  2. Line 566-567 mentions arterial aging as a clinical parameter – this term has no generally accepted units and measurement methods, unlike all the phenomena listed in these lines. Since this is a controversial term, and it is possible to judge vascular age only indirectly, I would recommend a more cautious formulation

  3. This scientific review provides an in-depth assessment of various types of wearable sensors, describes the physical principles underlying their application, the design features of sensors, their advantages and disadvantages. In a large number of cases, the information received with the help of wearable sensors is removed from the surface of the skin. The condition of the skin is certainly a key factor affecting the quality and accuracy of measurement of many parameters – in colorimetry, potentiometry, photoplethysmography, etc.

    It seems to me that it is necessary to make a fairly complete comment on the types and color of the skin (for example, according to the generally accepted classification of skin phototypes by Thomas B. Fitzpatrick routinely used in dermatology and cosmetology), its age features, structural features in different areas of the body – thickness, density of sweat glands (the latter is mentioned a little in line 781-782), blood supply, features chemical composition of the skin related to its barrier function – they affect the results obtained with the help of wearable biosensors, sometimes limiting their use, sometimes requiring the introduction of amendments to various properties and features of the structure of the skin.

Reviewer 3 Report

In this work, the authors summarize the transducer technologies used in wearable biosensors. The manuscript includes comprehensive description about all types of transducer technologies such as electrochemical, optical and thermal, and an outlook of factors for wearable sensor applications. Overall, this manuscript is well organized and written. However, there are two points regarding tables that need to be addressed by authors: 

  1. Table 2 (biomarker detection part) doesn’t include enough information. The authors should include more detailed description for each study such as the detection mechanism, limit of detection et al.
  2. Table 4, the authors should briefly summarize the advantages and shortcomings of each material and fabrication techniques
